# Attachment and Osteogenic Potential of Dental Pulp Stem Cells on Non-Thermal Plasma and UV Light Treated Titanium, Zirconia and Modified PEEK Surfaces

**DOI:** 10.3390/ma15062225

**Published:** 2022-03-17

**Authors:** Linna Guo, Ziang Zou, Ralf Smeets, Lan Kluwe, Philip Hartjen, Martin Gosau, Anders Henningsen

**Affiliations:** 1Department of Stomatology, The Second Xiangya Hospital, Central South University, Changsha 410011, China; 2Department of Oral and Maxillofacial Surgery, University Hospital Hamburg-Eppendorf, 20246 Hamburg, Germany; xiangya.zou@gmail.com (Z.Z.); r.smeets@uke.de (R.S.); kluwe@uke.de (L.K.); p.hartjen@uke.de (P.H.); m.gosau@uke.de (M.G.); 3Division Regenerative Orofacial Medicine, Department of Oral and Maxillofacial Surgery, University Hospital Hamburg-Eppendorf, 20246 Hamburg, Germany; a.henningsen@uke.de; 4Department of Gynecology and Obstetrics, The Third Xiangya Hospital, Central South University, Changsha 410013, China; 5Private Practice ELBE MKG, 22587 Hamburg, Germany

**Keywords:** ultraviolet light, non-thermal plasma, dental pulp stem cells, osteogenesis, titanium, zirconia, modified polyetheretherketone

## Abstract

Ultraviolet (UV) light and non-thermal plasma (NTP) treatment are chairside methods that can efficiently improve the biological aging of implant material surfaces caused by customary storage. However, the behaviors of stem cells on these treated surfaces of the implant are still unclear. This study aimed to investigate the effects of UV light and NTP treated surfaces of titanium, zirconia and modified polyetheretherketone (PEEK, BioHPP) on the attachment and osteogenic potential of human dental pulp stem cells (DPSCs) in vitro. Machined disks were treated using UV light and argon or oxygen NTP for 12 min each. Untreated disks were set as controls. DPSCs were cultured from the wisdom teeth of adults that gave informed consent. After 24 h of incubation, the attachment and viability of cells on surfaces were assessed. Cells were further osteogenically induced, alkaline phosphatase (ALP) activity was detected via a p-Nitrophenyl phosphate assay (day 14 and 21) and mineralization degree was measured using a Calcium Assay kit (day 21). UV light and NTP treated titanium, zirconia and BioHPP surfaces improved the early attachment and viability of DPSCs. ALP activity and mineralization degree of osteoinductive DPSCs were significantly increased on UV light and NTP treated surfaces of titanium, zirconia and also oxygen plasma treated Bio-HPP (*p* < 0.05). In conclusion, UV light and NTP treatments may improve the attachment of DPSCs on titanium, zirconia and BioHPP surfaces. Osteogenic differentiation of DPSCs can be enhanced on UV light and NTP treated surfaces of titanium and zirconia, as well as on oxygen plasma treated Bio-HPP.

## 1. Introduction

Dental implants are considered to be a common restorative therapy for patients with missing teeth. The clinical success of implant dentistry is closely related to rapid and predictable osseointegration [1]. After removal of cell debris directly after implant placement, newly synthesized bone tissue covers the whole implant surface, the implant is osteointegrated and can be able to support the load placed upon it. This process is carried out by mesenchymal stem cells (MSCs) [2]. After implant insertion, migratory MSCs are recruited to surgical sites and around the surfaces of the implants. The proliferation of these cells and differentiation into osteoblasts is an essential process for osseointegration in order to form the bone matrix around the implant insertion area. Bone formation around implants relies on the level of osteogenesis of surrounding MCSs, which can be largely affected by the physico-chemical surface characteristics of implants. Therefore, many strategies have been proposed to improve the surface characteristics of implants to promote bone formation at the bone-implant interface including modifications of the surface topography and chemical modifications [3,4,5]. For example, deposition of a calcium phosphates thin layer from supersaturated solutions in a mild condition is a relatively low-cost and easy implant coating method, which can provide a highly bioactive interface for surrounding bone tissue.

Compared with most modifications that require complex process procedures before being used in the clinical routine, ultraviolet light (UV) and non-thermal plasma (NTP) may have many advantages. It has been reported that four-week stored titanium surfaces may lead to a bisected ability of osseointegration and a significant reduction in bone-to-implant contact compared with newly prepared titanium implants [6]. In a previously published study, titanium and zirconia surfaces were treated with UV light and NTP using a short exposure time resulting in a significant increase of the wettability and oxygen content of surfaces and a significant decrease of carbon contents using either method, which may improve the impaired surface conditions of implant materials caused by long-time storage in customary packages [7,8,9]. UV light or NTP devices offer the possibility of practicable processing times and manageable size of the required devices, which may be favorable regarding the integration of these methods into the clinical routine. Therefore, UV light and NTP are considered as promising approaches to improve biocompatibility and the healing time of dental implant materials.

Titanium is mainly used as implant and abutment materials due to its excellent biocompatibility and mechanical properties. However, in the esthetic zone, titanium shows its limitation due to its dark grayish metallic color. In addition, as a metal device, its service life is affected by mechanical and environmental factors, such as fatigue, wear and fretting corrosion [10]. Thus, alternative materials have been introduced, for example, zirconia may be recommendable in esthetically challenging areas. It is reported that an all-ceramic superstructure can provide optimal esthetic results [11].

Currently, polymeric compounds, such as polyetheretherketone (PEEK), have been developed as additional substitutes for dental implants. However, compared with cortical bone, the elastic modulus of PEEK is very low [12]. A higher elastic modulus of PEEK is required for dental implant materials. Therefore, many reinforced PEEK materials have been developed, such as carbon or glass fiber-reinforced PEEK, which can be tailored to match the elastic modulus of cortical bone [13,14]. Recently, a ceramically enforced polyetheretherketone (PEEK) containing 25% ceramic fillers (with a grain size of 0.3–0.5 μm), named BioHPP, has gained more attention. Due to the small grain size, constant homogeneity can be achieved, which accounts for outstanding material properties [15]. Therefore, BioHPP has also been considered as promising dental implant material because of its elasticity, which is comparable to bone elasticity, its low plaque accumulation, low allergenic potential and no corrosion [16,17].

In a previous study, UV light, oxygen and argon plasma treatment significantly improved attachment of murine osteoblasts on titanium and zirconia surfaces [9]. Additionally, UV light and oxygen plasma treatment increased the attachment of fibroblast cells on titanium, zirconia and BioHPP surfaces [18]. However, the scientific knowledge about the effects of UV light and NTP treatment on osteogenic differentiation on these three materials is still lacking.

Dental pulp stem cells (DPSCs), a subgroup of which can be classified as multipotent MSCs, have gained wide attention in regenerative therapy resulting from their low costs, no known adverse effects on health and greater accessibility compared to the expensive and invasive techniques required for preparation other MSCs [19,20]. Particularly, DPSCs possess a high mineralization potential and may be regarded as beneficial for regenerative therapy of hard tissues [21,22]. Therefore, the investigation of the attachment and osteogenic potential of DPSCs have been examined in many studies assessing the effect of various treatments on implant materials [23,24]. However, the use of DPSCs on UV light and NTP treated titanium, zirconia and BioHPP surfaces have not been assessed and their ability for bone formation after surface treatment is not known.

The objective of this in vitro study was to assess the effect of UV irradiation and oxygen or argon based NTP treatment on the interaction between three types of implant materials (titanium, zirconia and BioHPP) and DPSCs concerning cell attachment and osteogenic potential. We bring forward a hypothesis that UV light and NTP treatment can improve the surfaces of the three types of implant materials and enhance attachment and osteogenic potential of DPSCs on these treated surfaces.

## 2. Materials and Methods

### 2.1. Sample Preparation

Samples of 15 mm diameter and 1.5 mm thickness were made from pure grade 4 titanium (Camlog, Basel, Switzerland). Zirconia disks were made from tetragonal zirconia polycrystal (ZrO_2_ 95%, Y_2_O_3_ 5%, 15 mm in diameter, 1.5 mm in thickness; Camlog, Basel, Switzerland) and PEEK disks were made from high-performance polyetheretherketone strengthened by 25% ceramic particles (BioHPP^®^, 15 mm in diameter, 1.5 mm in thickness; bredent GmbH, Senden, Germany). Sample roughness was analyzed using confocal microscopy, according to ISO 4288:1996. The mean arithmetic roughness (Ra) of titanium, zirconia and BioHPP samples was 1.8 μm, 1.4 μm and 1.5 μm, respectively. The sample preparation referred to our previous study [25]. All titanium, zirconia and BioHPP samples were sterilized and stored in customary packages for at least 4 weeks. Each sample was incubated in isopropanol for 30 min and washed with sterile distilled water three times and then dried in a sterile environment.

### 2.2. UV Light and NTP Treatment

Titanium, zirconia and BioHPP samples were randomly divided into one group of non-treated samples (controls) and three experimental groups. Disks in the first experimental group were treated using a UV light oven which generated UV light with an intensity of 0.15 mW/cm^2^ (λ = 253.7 nm). Disks in the other two groups were either treated with argon plasma or oxygen plasma using an NTP reactor (generator frequency 100 kHz, input power 24 W, system pressure 1 mbar, gas flow rate 1.25 sccm and gas purity > 99.5%, Diener Electronic GmbH, Ebhausen, Germany). All samples in the experimental groups were treated for 12 min.

### 2.3. Isolation and Culture of DPSCs

Cells were cultured from wisdom teeth of adults that were extracted at the Department of Oral and Maxillofacial Surgery, Eppendorf Medical Center Hamburg-Eppendorf, Germany. The outgrowth method was used to culture cells. Briefly, after soaking the teeth for 5 min in PBS, the teeth were broken using a hammer. Pulp tissues were carefully removed and soaked in modified Eagle medium (MEM, Gibco, Paisleg, UK) for 5 min. The tissues were cut into small pieces using tissue scissors, were put into 6-well plates and then cultured in 1 mL MEM with 15% fetal bovine serum (Gibco, Paisleg, UK) and 1% penicillin/streptomycin at 37 °C and 5% CO_2_. The culture medium was changed every three days. DPSCs were isolated using a low-density seeding method [26] and identified via successful multi-differentiation.

All methods were performed in accordance with the relevant guidelines and regulations (University Hamburg-Eppendorf). The specimens were recovered from bio-waste, which does not require ethical approval according to the local regulation in Hamburg, Germany. However, the study protocols were registered in the Hamburg Privacy Protection Office. All samples were completely anonymized according to the local privacy protection regulation. The study protocol was reported to the corresponding local authority. All patients signed written informed consent forms for using their anonymized waste specimen scientifically.

### 2.4. Cell Attachment and Morphology

Confocal laser scanning microscopy (TCS SP8 X, Leica Microsystems, Wetzlar, Germany) and fluorescence microscopy (Zeiss Axiovert M200, Helmholtz Zentrum, Neuherberg, Germany) were used to assess cell attachment and morphology using 60-fold and 20-fold objective lenses, respectively. After 24 h of incubation and 21 days of osteogenic induction, cells were fixed using 4% paraformaldehyde for 30 min and permeabilized with 0.1% Triton X-100/PBS (Gibco, Invitrogen, Paisley, UK) for 15 min at room temperature. After rinsing three times using PBS, F-actin filaments were stained using a fluorescent dye (biotinylated phalloidin, Alexa Fluor 488 green, 1:1000; Thermo Fisher Scientific, Waltham, MA, USA) and incubated for 60 min at room temperature. After that, samples were washed with PBS three times and dried in normal air. Antifade Mountant (Fluoromount-G, Southern Biotech, Birmingham, AL, USA) was used to fix all samples on glass-bottom dishes (WillCo-Dish, Amsterdam, The Netherlands) and they were stored in the dark at 4 °C.

### 2.5. Viability Assay

After 24 h of incubation, the viability of cells was assessed using a CellTiter 96^®^ Aqueous Non-Radioactive Cell Proliferation Assay Kit (MTS assay, Promega, Madison, WI, USA). Briefly, 20% MTS solution was added to each well and the plates were incubated for 1–4 h at 37 °C in a humidified, 5% CO_2_ atmosphere. The absorbance was measured using a microplate reader at a wavelength of 490 nm.

### 2.6. Osteogenic Differentiation

DPSCs osteogenic differentiation was carried out according to the previously reported procedure. The differentiation medium was prepared using DMEM/Hams F12 with 10% human serum supplemented with 100 nM dexamethasone, 50 µM ascorbic acid 2-phosphate and 10 mM β-glycerophosphate (Sigma-Aldrich, St. Louis, MO, USA). The media was exchanged every second day and the differentiation was continued for three weeks.

### 2.7. Alkaline Phosphatase Activity (ALP)

Osteogenic differentiation was additionally quantified by measuring ALP activity using a p-Nitrophenyl phosphate assay (pNPP) following the manufacturer’s protocol after 14 and 21 days of incubation [27]. pNPP is able to conjugate with ALP and form a soluble end product in yellowish color and can be read out spectrophotometrically at 405 nm. The enzyme activity values were normalized against the viability of the cells.

### 2.8. Mineralization Assay

After 21 days of culture in osteogenic medium, calcium levels were assessed by the colorimetric method using a Calcium Assay kit (Cayman Chemical, Ann Arbor, MI, USA). The concentration of calcium was estimated using a spectrophotometer and calculated by measuring optical density at 570 nm. All values were normalized against the viability of the cells.

### 2.9. Statistical Analysis

Statistical analysis was performed using SPSS 21 (IBM, Armonk, NY, USA). Normal distribution of viability, ALP and mineralization values were assessed using the skewness-kurtosis method. Afterward, all values were compared using one-way analysis of variance (ANOVA). For all results, statistical significance of differences was set at *p* < 0.05.

## 3. Results

### 3.1. Cell Attachment and Morphology

Qualitative observation of immunofluorescent labeled cells showed that cells on UV light, argon and oxygen plasma treated surfaces were generally larger and more elongated after 24 h of culture compared to control groups. The actin cytoskeleton was marked with phalloidin (green color) and the nucleus by DAPI (4′,6-diamidino-2-phenylindole, blue color). In untreated groups of titanium, zirconia and BioHPP, small and hardly attached cells were observed (Figure 1A,E,I). However, on all UV light or NTP treated surfaces of three types of materials, cells showed a flattened and spread cytoskeleton with long cytoplasmic extensions (Figure 1B–D,F–H,J–L).

Figure 2 shows the attachment of osteoinductive cells at 21 days of culture. On untreated surfaces of titanium, zirconia and BioHPP, adhesion of cells was weak and loosely distributed (Figure 2A,E,I). Either UV light or NTP treatment resulted in strong cell adhesion and dense distribution (Figure 2B–D,F–H,J–L), whereas compared with the other two experimental groups, argon plasma treatment allowed only sparse cell adhesion (Figure 2D,H,L).

### 3.2. Viability

Cell viability in each group after 24 h of incubation is shown in Figure 3. Compared with controls, the viability of DPSCs on UV light, oxygen and argon plasma treated surfaces of titanium, zirconia and BioHPP significantly increased (*p* < 0.05). However, there were no significant differences in the results between the treatments.

### 3.3. Differentiation of DPSCs on Treated Materials

Osteogenic differentiation was investigated by analyzing ALP activity (day 14 and 21) and the degree of mineralization (day 21) under osteogenic induction conditions. ALP activity was generally increased after UV light and NTP treatment compared to control groups (Figure 4). Except for the argon treated surface of titanium at day 14 and BioHPP at day 21, the increase in ALP between other treatment groups and their corresponding control group was statistically different.

Mineralization was quantified at day 21 and is shown in Figure 5. Although any surface treatment led to increased mineralization on all types of disks, differences were only significant after UV light and oxygen plasma treatment on all three types of materials and for argon plasma treatment on zirconia (*p* < 0.05).

## 4. Discussion

Titanium, zirconia and BioHPP are currently the most widely used dental implant and dental implant abutment materials. To the authors’ knowledge, no previous study has investigated and compared the effects of UV light, non-thermal oxygen and argon plasma treatment on attachment and osteogenic differentiation of DPSCs on these materials in order to assess the effect of these treatments from the perspective of bone regeneration. The results of this study revealed that UV light and non-thermal oxygen and argon plasma treatment may facilitate the early attachment of DPSCs and also adhesion of osteoinductive DPSCs. Compared to non-treated surfaces, osteogenic activity of DPSCs was significantly increased on UV light and oxygen plasma treated surfaces on all three types of materials (*p* < 0.05). Argon plasma treatment had only small effects on titanium and BioHPP disks regarding osteogenesis after 21 days, however, it significantly increased degrees of osteogenic differentiation on titanium and zirconia disks.

The main limitation of the current study is that all experiments were carried out in vitro. Additionally, using two-dimensional static cell culture may be another limitation of this study, since static cell culture may ignore the influence of cell hydrodynamics in an in vivo environment. Therefore, further in vivo studies are needed to verify the results of this in vitro study and to put the results of this study into bigger contexts.

DPSCs from permanent teeth with a neural crest origin are a subgroup of MSCs, which have been proven to be able to differentiate into bone producing osteoblastic cells efficiently [28,29]. Compared with other MSCs, such as bone marrow stem cells, DPSCs have been proven to be a more effective resource to improve osseointegration on the surfaces of titanium in vivo [30,31]. Despite their obvious osteogenic properties, the application of DPSC in dental implant research as well as in clinical trials is limited so far. DPSCs could be used for a personalized therapy from the perspective of improving dental implant osseointegration because the removed teeth could provide a very valuable origin of autologous DPSCs [32].

Titanium and zirconia are most frequently used as materials for dental implants, while PEEK has gained more attention in orthopedic research over recent years. Some researchers found that PEEK can be well applied in the field of oral implants. For example, PEEK can be used as an implant abutment which can exhibit the same survival rate and biologic and esthetic outcomes as zirconia abutment at the five-year evaluation [33]. However, because of its limited osteogenic properties, various studies are related to the increase of the biological activity of PEEK implants [34]. Nakonieczny et al. revealed that zirconia can be used with polymer to prepare composites for biomedical applications such as FDM filament for 3D printing [35]. Mishra et al. found that zirconia particles in PEEK matrix could significantly improve the storage modulus of the nanocomposites [36]. Recently, BioHPP as a modified PEEK strengthened using ceramic filler material was created and optimized for dental use due to its excellent material characteristics [37,38]. From the perspective of mechanical characteristics, BioHPP possesses a level of elasticity which is very close to human spongious bone, leading to the reduction of the risk of fracture, and it may be favorable for implant-supported prosthetic solutions [16]. However, research regarding surface modification of this emerging and promising dental material is still lacking. In this study, titanium, zirconia and BioHPP were used to evaluate and compare the effect of surface treatments on attachment and osteogenic differentiation of DPSCs.

As a component of direct functional and structural connection between living bone and implant surface, implant design and surface treatment play an important role in osseointegration [39]. Some studies revealed recently that chairside devices producing UV light and NTP which can be applied immediately prior to implant placement are able to reduce carbon pollution and may increase the number of hydroxyl groups, resulting in surface hydrophilization and enhancing the interactions between biomaterials and cells [40,41,42]. In addition, UV irradiation has been proven to enhance cellular adhesion and protein adsorption by transforming the electrostatic state of the surfaces of materials into positive charge [43]. NTP is able to reduce the formation of bacterial colonies in vitro as well as to improve cytocompatibility on alloplastic materials [44,45]. Previously published results revealed that UV light and NTP treatment could enhance the attachment of various cells such as murine fibroblast (L929) and human fibroblasts on titanium, zirconia and BioHPP surfaces and also the attachment of murine osteoblast-like cells MC3T3 on machined titanium surfaces [7,18]. In another of our studies, the viability and cellular expression of MC3T3-E1 cells as well as their expression of vascular endothelial growth factor (VEGF) on titanium and zirconia surfaces was significantly increased after 12 min UV light or 1 min NTP treatment [25]. In the present study, UV light, oxygen and argon plasma treatment also increased the adhesion of DPSCs after 24 h of incubation. Hersel et al. reported that attachment, spreading, cytoskeleton development and formation of cell-matrix adhesions are a cascade of molecular events of initial cell–biomaterial interaction [46]. During cell spreading and focal adhesion formation, survival and proliferation of anchorage dependent cells can be activated [47]. Generally, the subsequent function of such anchorage dependent cells can be regulated by the nature and degree of such initial cell–biomaterial interactions [48,49]. Therefore, initial settlement and retention of DPSCs may be crucial processes for further osteogenic differentiation.

McBeath et al. demonstrated that cell shape can regulate the commitment of MSCs in terms of osteogenesis, which were allowed to adhere, flatten and spread, while unspread and round cells became adipocytes [50]. In this study, the results showed that after 21 days of osteogenic induction, compared with control groups, cells on UV light and NTP treated surfaces of the three types of materials stretched better, flatter and more numerously, indicating that osteogenesis of DPSCs may be enhanced by these treatments. However, cell reactions on argon plasma treated surfaces were not as good as in the other two experimental groups, but they were at least better than in the control group. ALP activity and mineralization degree were evaluated in each group in order to quantify the osteogenesis of DPSCs further. All values of ALP and mineralization were normalized against their responding cell viability values to suppress the effect of inconsistent cell numbers in each group. The results showed that ALP activity was significantly increased after UV light and NTP treatment except for the argon treated surface of BioHPP compared to control groups. Any surface treatment led to increased mineralization degrees on titanium, zirconia and BioHPP disks. However, differences were only significant after UV light and NTP oxygen treatment on all three types of materials and after NTP argon treatment on zirconia.

These results demonstrate that the osteogenesis of DPSCs may be improved after UV light and oxygen plasma treatment of titanium, zirconia and BioHPP surfaces and also on argon plasma treated titanium and zirconia surfaces. Zhang et al. found that UV light treated titanium enhanced the osteogenic activity of rat bone marrow mesenchymal stem cells as indicated by increased levels of adhesion, osteogenic factor production, alkaline phosphatase activity and osteogenesis-related gene expression, which is consistent with the results of this study [51]. Althaus et al. showed that adipose tissue-derived mesenchymal stem cells exhibited a doubled mineralization degree on oxygen and ammonia (10 and 50 W) plasma treated PEEK surfaces compared to the untreated PEEK surfaces [52]. However, under the conditions of this experimental setting, argon plasma treatment increased the attachment and viability of DPCSs on all three types of materials, but the effect of this treatment on the osteogenic activity of DPSCs on modified PEEK (BioHPP) surfaces was not significant. This may be related to the wattage of the NTP device used in this study (input power 24 W), the time of treatment, the cell type or the material structure, which should be explored further. Additionally, it has to be mentioned that this study is only an in vitro study. Implications of the determined effects for clinical application should to be evaluated in further studies.

## 5. Conclusions

The aim of the study was to determine the influence of UV irradiation and oxygen or argon based NTP treatment on the interaction between three types of implant materials (titanium, zirconia and BioHPP) and DPSCs concerning cell attachment and osteogenic potential. Generally, surface treatment using UV light, non-thermal oxygen and argon plasma may improve the attachment and viability of DPSCs on titanium, zirconia and ceramic reinforced PEEK surfaces of dental implant materials. Osteogenesis of DPSCs may be enhanced on UV light and non-thermal oxygen plasma treated surfaces of titanium, zirconia and BioHPP and also on non-thermal argon plasma treated zirconia surfaces, whereas the effect of argon plasma treatment on titanium and BioHPP may be inferior to the other two methods. The results of the present study provide a theoretical basis for the applications of UV light and NTP treatment to enhance the osteoinductive effects of treated implant surfaces. However, due to the limited validity of the in vitro results, it is necessary to conduct in vivo studies to verify them.

## Figures and Tables

**Figure 1 materials-15-02225-f001:**
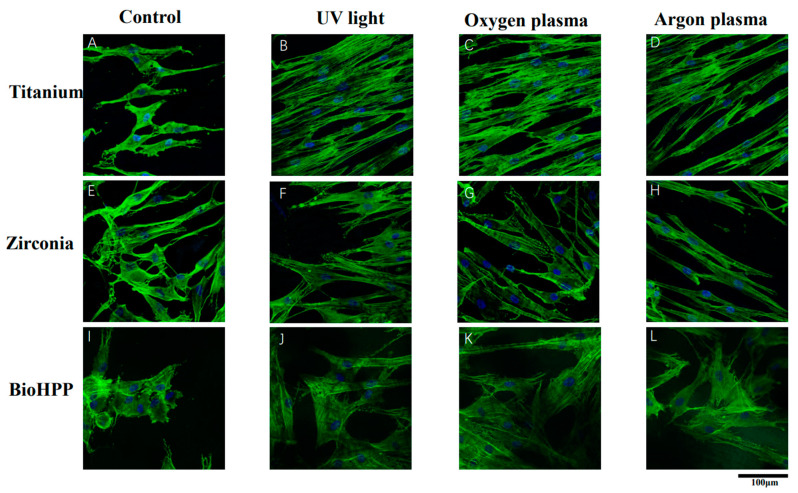
Cell attachment and morphology on different surface treated titanium, zirconia and BioHPP after 24 h. Representative examples of cytoskeletons stained with phalloidin after 24 h of incubation on controls (**A**,**E**,**I**), UV light treated (**B**,**F**,**J**), oxygen plasma treated (**C**,**G**,**K**) and argon plasma treated (**D**,**H**,**L**) surfaces of titanium, zirconia and BioHPP using confocal microscopy with 60-fold objective lens. The actin cytoskeleton was marked phalloidin (green color) and the nucleus by DAPI (blue color). Compared to control groups, cells were more extended and flattened with a more widely spread cytoskeleton possessing elongated cytoplasmic extensions on UV light, oxygen and argon plasma treated surfaces.

**Figure 2 materials-15-02225-f002:**
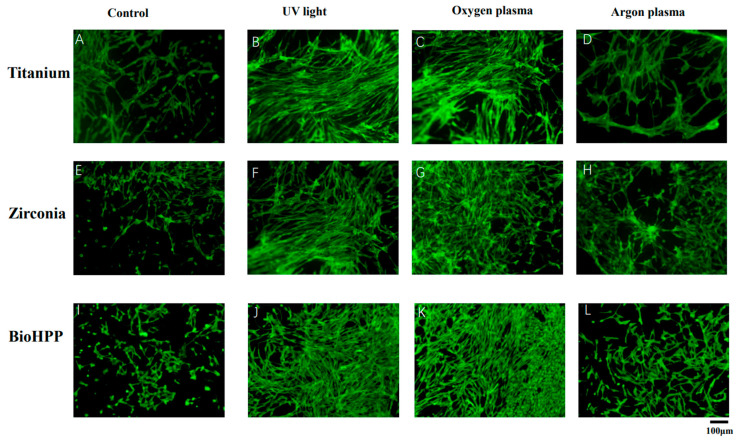
Cell attachment and morphology of osteoinductive DPSCs on different surface treated titanium, zirconia and BioHPP after 21 days. The attachment of osteoinductive cells at 21 days of culture was observed using fluorescence microscopy (20×). Adhesion of cells was weak and cells were loosely distributed on untreated surfaces of titanium, zirconia and BioHPP (**A**,**E**,**I**). Either UV light or NTP treatment resulted in stronger cell adhesion and dense distribution (**B**–**D**,**F**–**H**,**J**–**L**), whereas argon plasma treatment led only to sparse cell adhesion (**D**,**H**,**L**).

**Figure 3 materials-15-02225-f003:**
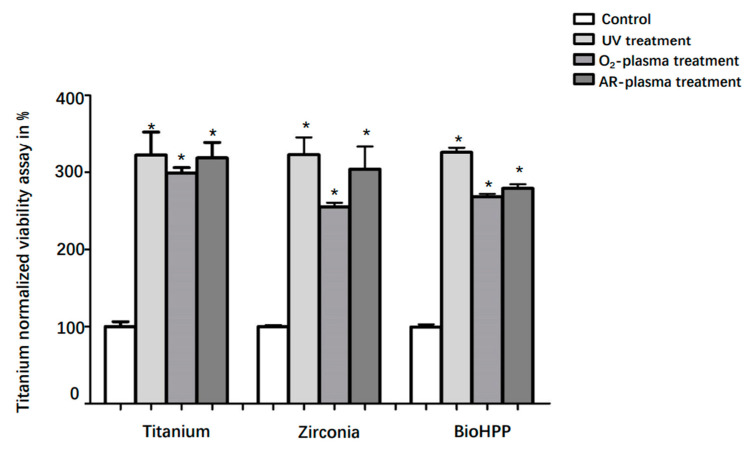
Viability of DPSCs on different surface treated titanium, zirconia and BioHPP after 24 h. Viability of DPSCs on controls and surface treated titanium, zirconia and BioHPP disks after 48 h of incubation was shown. Compared to controls, the viability of DPSCs on UV light, oxygen and argon plasma treated surfaces of titanium, zirconia and BioHPP was significantly increased (*p* < 0.05). There were no significant differences between the treatments. * *p* < 0.05.

**Figure 4 materials-15-02225-f004:**
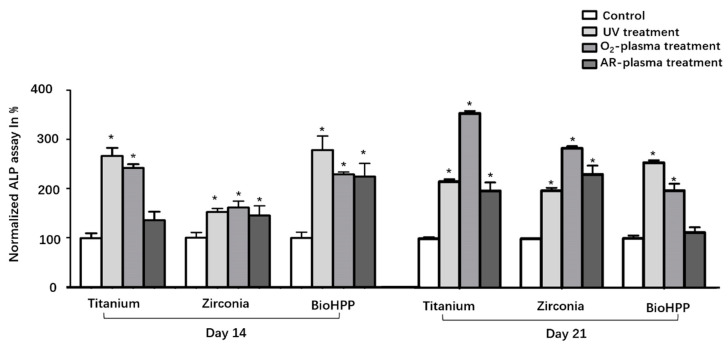
ALP activity of osteoinductive DPSCs on different surface treated titanium, zirconia and BioHPP after 14 and 21 days. ALP activity was increased in general after UV light and NTP treatment compared to control groups. The increase in ALP between the experimental groups and their corresponding control group was statistically significant except for the argon treated surface of titanium at 14 days and BioHPP at 21 days. * *p* < 0.05.

**Figure 5 materials-15-02225-f005:**
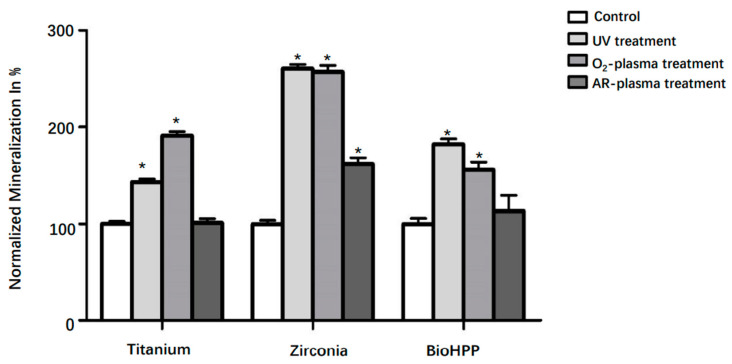
Mineralization of osteoinductive DPSCs on different surface treated titanium, zirconia and BioHPP after 21 days. Although any surface treatment led to increased mineralization on titanium, zirconia and BioHPP disks, differences were only significant after UV light and oxygen treatment on all three types of materials and argon treatment on zirconia (*p* < 0.05). * *p* < 0.05.

## Data Availability

The data that support the findings of this study are available within the manuscript. Additional information can be provided from the corresponding author upon reasonable request.

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
