# Peer review of "Attachment and Osteogenic Potential of Dental Pulp Stem Cells on Non-Thermal Plasma and UV Light Treated Titanium, Zirconia and Modified PEEK Surfaces"

_materials, 2022, doi:10.3390/ma15062225_

Round 1

Reviewer 1 Report

Dear authors,

Your article is very good as far as it goes. It is very interesting and up-to-date because of the consideration of modified PEEK. In principle, I have no major comments. The following describes the strengths and weaknesses of your article. Please correct any minor remarks and then I recommend the article for acceptance.

STRONG POINTS

- actual subject matter,
- very well described research methodology,
- interesting research with valuable conclusions,
- graphically very well designed article.

WEEK POINTS

-lack of a "typical" material method, e.g. SEM or simple optical microscopy to show the surface of connectors,
- as you are dealing with modified PEEK, it would be worthwhile to think about the current works on this subject in dentistry, especially the modification of PEEK and its application in additive methods, e.g. FDM; please read and supplement the literature with the following works:

(1)Randomized clinical trial of zirconia and polyetheretherketone implant abutments for single-tooth implant restorations: A 5-year evaluation

(2) Alumina and Zirconia-Reinforced Polyamide PA-12 Composites for Biomedical Additive Manufacturing 

(3) PEEK composites reinforced with zirconia nanofiller

- The article has a few editing errors: e.g. lower indices of chemical formulae and formatting of literature; please go through the whole article again and correct these minor errors.

Thank you for your article, it is valuable and will certainly achieve good citability. Super work.

Regards

Reviewer

Author Response

  1. Lack of a "typical" material method, e.g. SEM or simple optical microscopy to show the surface of connectors.

Response: Thank you for your suggestion. As you said, we did ignore the description of the typical material method. We added some descriptions concerning the sample roughness analysis by confocal microscopy in the method section (line120-122, page 3). According to your valuable suggestions, more parameters about the surface of connectors will be detected in our further researches.

  1. As you are dealing with modified PEEK, it would be worthwhile to think about the current works on this subject in dentistry, especially the modification of PEEK and its application in additive methods, e.g. FDM; please read and supplement the literature with the following works:

        (1)Randomized clinical trial of zirconia and polyetheretherketone implant abutments for single-tooth implant restorations: A 5-year evaluation

         (2) Alumina and Zirconia-Reinforced Polyamide PA-12 Composites for Biomedical Additive Manufacturing 

         (3) PEEK composites reinforced with zirconia nanofiller

Response: Thank you for your suggestion. I read the three articles carefully and found that there are many valuable clues that can be mentioned in the manuscript. We especially mentioned the modification of PEEK and its application in additive methods in the discussion section (line 288-296, page 8-9). We also cited these three very valuable articles.

  1. The article has a few editing errors: e.g. lower indices of chemical formulae and formatting of literature; please go through the whole article again and correct these minor errors.

Response: Thank you for your suggestion. We went through the whole article again and revised number of typos and grammatical errors in the manuscript accordingly (line 130, 143, 172, 180, 186, 192, 265 and 319). We checked the correctness of literary references according to the requirements of MDPI journals carefully. We revised some errors in reference section.

Reviewer 2 Report

The article entitled “Attachment and osteogenic potential of dental pulp stem cells on non-thermal plasma and UV light treated Titanium, Zirconia and modified PEEK surfaces” aimed to investigate

the effects of UV light and NTP treated surfaces of titanium, zirconia and modified polyetheretherketone (PEEK, BioHPP) on human dental pulp stem cells (DPSCs) in vitro. Authors have well revised several issues; however, I ask authors to add some key concepts. 

  •  The abstract is confusing, please write the entire paragraph
  • The introduction section needs to be better reorganized; authors should include studies regarding tools for bone morphology analysis (please, see and discuss doi: 10.1002/term.2494.)
  • It would be interesting to focus as underlying titanium abutment can reduce the porcelain translucency, resulting in a darkened prosthesis (please, see and discuss PMID: 21359243) and the use of zirconia in daily practice can provide harmonious gingival architecture around implant restorations, improving both the biologic and physiologic aspects.
  • In fact, the authors must offer a general overview of the pros and cons of titanium and zirconia (used in implant fixtures, abutments, and crowns) and point out the cause of the survival rates of titanium implants
  • Is there a null hypothesis of the study? Please describe paper’s aim better,
  • The part of materials and methods is well described,
  • Are there any references regarding the preparation of the study samples?

Minor issues:

  • Conclusions cannot be reduced to a sentence: you must improve them highlighting the limits and the future insights pointed out from this article.
  • Several moderate typos are present in the text, please, amend
  • The references must be reformatted according to the instructions for the authors (see Journal Articles: Author 1, A.B.; Author 2, C.D. Title of the article. Abbreviated Journal NameYearVolume, page range.)

According to this Reviewer’s consideration, novelty and quality of the paper, publication of the present manuscript is recommended after major revision.

Author Response

Thank you for your suggestion. I appreciate your carefully reading and valuable suggestions. We also added some key concepts concerning reinforced PEEK in the manuscript (line 79-84, page 2).

Reviewer 3 Report

Peer-reviewing the article (materials-1576289):

“Attachment and osteogenic potential of dental pulp stem cells on non-thermal plasma and UV light treated Titanium, Zirconia and modified PEEK surfaces” in the MDPI Materials journal.

The peer-reviewed article is written scientifically competently and made only positive impressions on me. The manuscript is quite easy to read, the relevance of the study is justified and the obtained results are clearly understandable. However, after careful reading, I have a few comments:

  • The text of the article contains a sufficient number of typos and grammatical errors, the indexes are not put down properly everywhere. I have noted some points in the text. Please correct the general style of the text.
  • Figure 5 is clearly in the wrong place, and the sub-chapter between 2.6 and 2.7 is also probably missing.
  • Why was 21 days allocated for tracking osteogenic processes? Is there a need to monitor the dynamics of processes for a longer period of time?
  • What determines the choice of these types of material for implants and types of their treatments? In my opinion, in the "Introduction" it will be interesting to read a small comparative analysis of the advantages and disadvantages with materials based on calcium phosphates, as one of the promising compounds for orthopedics and dentistry.
  • As you indicated in the text, "in vivo" studies are needed to fully understand all osteoregenerative processes. What are your prospects for conducting these studies "in vivo" in the future?
  • In my opinion, you need to supplement or expand the section "Сonclusion" to increase the significance of your work.
  • Please check the correctness of the design of literary references according to the requirements of MDPI journals.

Author Response

Response to Reviewer 3:

  1. The text of the article contains a sufficient number of typos and grammatical errors, the indexes are not put down properly everywhere. I have noted some points in the text. Please correct the general style of the text.

Response: Thank you for your suggestion. We revised number of typos and grammatical errors in the manuscript which are marked by you (line 130, 143, 172, 180, 186, 192, 265 and 319). In addition, we carefully check the whole manuscript and correct some other errors.

  1. Figure 5 is clearly in the wrong place, and the sub-chapter between 2.6 and 2.7 is also probably missing.

Response: Thank you for your suggestion. We have corrected them accordingly (line 180, page 4). We renumbered figures. Figure 5 should be numbered as Figure 2 (line 220, page 6).

  1. Why was 21 days allocated for tracking osteogenic processes? Is there a need to monitor the dynamics of processes for a longer period of time?

Response: Thank you for your suggestion. Jensen et al. investigated osteogenic differentiation of bone marrow-derived mesenchymal stem cells (BMSCs) and dental pulp-derived stem cells (DPSCs) in vitro and in a pig calvarial critical-size bone defect model. The results of the study revealed that DPSCs exhibited a higher osteogenic potential compared with BMSCs [1]. It is reported that early-stage osteogenic genes such as RUNX2, collagen type I, and ALP were expressed at day 14 in osteogenic differentiating BMSCs. And after 21 days of osteogenic induction, BMSCs had well mineralization [2]. Therefore, early-stage osteogenesis and mineralization of DPSCs can be detected after 14 and 21 days of osteogenic induction respectively. However, it is an interesting idea to monitor the dynamics of the processes for a longer period of time. We will explore it in future research.

  1. Jensen, J.; Tvedesøe, C., et al., Dental pulp-derived stromal cells exhibit a higher osteogenic potency than bone marrow-derived stromal cells in vitro and in a porcine critical-size bone defect model. Sicot-J 2016, 2, 16.
  2. Mohamed-Ahmed, S.; Fristad, I., et al., Adipose-derived and bone marrow mesenchymal stem cells: a donor-matched comparison. Stem Cell Res Ther 2018, 9, (1), 168.

  1. What determines the choice of these types of material for implants and types of their treatments? In my opinion, in the "Introduction" it will be interesting to read a small comparative analysis of the advantages and disadvantages with materials based on calcium phosphates, as one of the promising compounds for orthopedics and dentistry.

Response: Thank you for your suggestion. Titanium is commonly used as implant materials in the daily dental clinic practice. Zirconia implant is recommendable used in esthetic areas. And BioHPP has also been considered as a promising dental implant material due to its elasticity which is comparable to bone elasticity. All the three of materials in the present study have been allowed to be used as implant materials in dental clinic. As you said, in recent years, the design and development of novel materials for biomineralized tissues is an extremely attractive field of research where calcium phosphates-based materials applications play a leading role. However, because of its porous structure and poor mechanical properties, calcium phosphates cannot be directly used as an implant material. While it can be applied as a method of surface coating of implant to promote osseointegration between bone and implant interfaces [3]. So, we added a small comparative analysis of the advantages and disadvantages with materials based on calcium phosphates in introduction section (line 54-59, page 2) accordingly. The treatment (UV-light and non-thermal plasma) in the present study is mainly aimed at dealing with the problem of biological aging on the surface of implant materials stored in the customary package. Either of them can be used as a chair-side treatment with simple processed procedures in dental clinic.

  1. Bigi, A.; Boanini, E., Functionalized biomimetic calcium phosphates for bone tissue repair. J Appl Biomater Funct Mater 2017, 15, (4), e313-e325.
  2. As you indicated in the text, "in vivo" studies are needed to fully understand all osteoregenerative processes. What are your prospects for conducting these studies "in vivo" in the future?

Response: Thank you for your question about the in vivo study.  The in vivo study will aim to determine the influence of UV-light and non-thermal plasma on osteointegration of titanium, zirconia and BioHPP implants. All implant materials need to be stored in customary package for 4 weeks. Then, Implants will be treated with UV-light or non-thermal plasma for 12min and inserted into the parietal bones of domestic pigs. The animals were sacrificed after a healing interval of 4 and 8 weeks. The degree of osseointegration was determined using histomorphometric determination of bone-to-implant contact values (BIC) and the bone-to-implant contact values within the retentive parts of the implants (BAFO).

  1. In my opinion, you need to supplement or expand the section "Сonclusion" to increase the significance of your work.

Response: Thank you for your suggestion. We expand the conclusion section to increase the significance of our work (line 366-374, page 10).

  1. Please check the correctness of the design of literary references according to the requirements of MDPI journals.

Response: Thank you for your suggestion. We checked the correctness of the design of literary references according to the requirements of MDPI journals carefully. We revised some errors in reference section accordingly.

Reviewer 4 Report

In this study, the authors have studied the osteogenic potential of human dental pulp stem cells using PEEK disc. The PEEK surface was modified by titanium, zirconia and BioHPP by treating UV and non-thermal plasma. The disc was characterized alkaline phosphatase activity and mineralization degree and by Calcium Assay. The authors concluded that the UV and non-thermal treated disc showed better osteogenic differentiation of cells.  However detailed characterization may improve the quality of the study. In the current reviewer’s opinion, the manuscript may be considered for a major revision.

Comments to the Authors

  1. In the introduction, authors may give more details of PEEK modified metals for dental applications.
  2. The authors may give schematic diagram for the preparation of PEEK disk treated with titania, Zirconia and BioHPP.
  3. The authors may provide other characterization PEEK disk such as EDX, DSC and TGA to confirm the functional and its degradation property.
  4. Authors may mention method used for the sterilization of the disk.
  5. Authors may used commercially available material to compare disk for the osteogenic differentiation of cells.
  6. Alkaline phosphatase is early marker for osteogenic differentiation of cells. Author may have studied the earlier time such as day 3 and day 7.

Author Response

Response to Reviewer 4:

  1. -In the introduction, authors may give more details of PEEK modified metals for dental applications.

        -The authors may give schematic diagram for the preparation of PEEK disk treated with titania, Zirconia and BioHPP.

       -The authors may provide other characterization PEEK disk such as EDX, DSC and TGA to confirm the functional and its degradation property.

Response: Thank you for your suggestion. The preparation of PEEK disk treated with metal materials is really an interesting idea, however, in the present study, we didn’t treat metal with PEEK. We used a promising implant material BioHPP which is a PEEK containing 25% ceramic fillers.

-Sorry for the unclear, we rewrote the description of BioHPP in introduction section (line 83-86, page 2). We also added more details of modified PEEK for dental application in introduction section (line 84-85, page 2).

-We make a schematic diagram for the preparation of our treated disks (Supplementary figure 1).

-According to your suggestion, we also mentioned other metal-free reinforced PEEK to further elucidate the function of this kind of materials (line 81-83, page 2).

  1. -Authors may mention method used for the sterilization of the disk.

        -Authors may used commercially available material to compare disk for the osteogenic differentiation of cells.

Response: Thank you for your suggestion.

-We mentioned the method concerning sterilization of the disk in method section accordingly (line123-125, page 3).

-Titanium and zirconia are commercially available implant materials which have been commonly used in dental implant [4]. In the present study, we compare the osteogenic differentiation of cells on modified PEEK with commercially used implant material titanium and zirconia.

Koller, M.; Steyer, E.; Theisen, K.; Stagnell, S.; Jakse, N.; Payer, M., Two-piece zirconia versus titanium implants after 80 months: Clinical outcomes from a prospective randomized pilot trial. Clin Oral Implants Res 2020, 31, (4), 388-396.

  1. Alkaline phosphatase is early marker for osteogenic differentiation of cells. Author may have studied the earlier time such as day 3 and day 7.

Response: Thank you for your suggestion. We actually detected the value of ALP after 14 days of osteogenic induction. Because the trend was the same as that on day 21, it was not showed in the results. But I agree with you that ALP is an early marker for osteogenic differentiation of cells. It is more convincing to display it. Therefore, we added the results related descriptions of ALP value after 14 days osteogenic induction in results section ( Figure 4 and line237-242, page 7).

Round 2

Reviewer 2 Report

The authors adequately addressed the suggestions of this reviewer

Reviewer 4 Report

Manuscript may be accepted.

This manuscript is a resubmission of an earlier submission. The following is a list of the peer review reports and author responses from that submission.

Round 1

Reviewer 1 Report

The article entitled “Attachment and osteogenic potential of dental pulp stem cells on non-thermal plasma and UV light treated Titanium, Zirconia and modified PEEK surfaces” aimed to investigate

the effects of UV light and NTP treated surfaces of titanium, zirconia and modified polyetheretherketone (PEEK, BioHPP) on human dental pulp stem cells (DPSCs) in vitro. Authors have well revised several issues; however, I ask authors to add some key concepts. 

  •  The abstract is confusing, please write the entire paragraph
  • The introduction section needs to be better reorganized; authors should include studies regarding the failure of titanium implants (please, see and discuss doi:10.1177/03946320110240S214)
  • It would be interesting to focus as underlying titanium abutment can reduce the porcelain translucency, resulting in a darkened prosthesis (please, see Traini T, Pettinicchio M, Murmura G, et al. Esthetic outcome of an immediately placed maxillary anterior single-tooth implant restored with a custom -made zirconia-ceramic abutment and crown: a staged treatment. Quintessence Int. 2011; 42 (2): 103-108.) and the use of zirconia in daily practice can provide harmonious gingival architecture around implant restorations, improving both the biologic and physiologic aspects.
  • In fact, the authors must offer a general overview of the pros and cons of titanium and zirconia (used in implant fixtures, abutments, and crowns) and point out the cause of the survival rates of titanium implants
  • Is there a null hypothesis of the study? Please describe paper’s aim better,
  • The part of materials and methods is well described,
  • Are there any references regarding the preparation of the study samples?

Minor issues:

  • Conclusions cannot be reduced to a sentence: you must improve them highlighting the limits and the future insights pointed out from this article.
  • Several moderate typos are present in the text, please, amend
  • The references must be reformatted according to the instructions for the authors (see Journal Articles: Author 1, A.B.; Author 2, C.D. Title of the article. Abbreviated Journal NameYearVolume, page range.)

According to this Reviewer’s consideration, novelty and quality of the paper, publication of the present manuscript is recommended after major revision.

Reviewer 2 Report

Dear authors,

I have read your article in depth. Unfortunately, it is not a publication about materials research or materials engineering in general. In my opinion, you made a mistake with the journal you sent it to - the article is more clinical and in my opinion should have been sent to journals such as:

- https://www.mdpi.com/journal/jcm/special_issues/clinical_dental

I am therefore very sorry but I have to reject this article because it does not fit in with the profile of the magazine. I wish you more luck with your clinical journals

Best Wishes

Reviewer